

# Investigating mortality salience as a potential causal influence and moderator of responses to laboratory pain

Beibei You[1,2], Hongwei Wen[2] and Todd Jackson[3]

[1] School of Nursing, Guizhou Medical University, Guiyang, China
[2] Faculty of Psychology, Southwest University, Chongqing, China
[3] Department of Psychology, University of Macau, Taipa, China

## ABSTRACT

**Background:** Because pain can have profound ramifications for quality of life and daily functioning, understanding nuances in the interplay of psychosocial experiences with pain perception is vital for effective pain management. In separate lines of research, pain resilience and mortality salience have emerged as potentially important psychological correlates of reduced pain severity and increased tolerance of pain. However, to date, there has been a paucity of research examining potentially interactive effects of these factors on pain perception. To address this gap, the present experiment investigated mortality salience as a causal influence on tolerance of laboratory pain and a moderator of associations between pain resilience and pain tolerance within a Chinese sample.

**Methods:** Participants were healthy young Chinese adults (86 women, 84 men) who first completed a brief initial cold pressor test (CPT) followed by measures of demographics and pain resilience. Subsequently, participants randomly assigned to a mortality salience (MS) condition completed two open-ended essay questions in which they wrote about their death as well as a death anxiety scale while those randomly assigned to a control condition completed analogous tasks about watching television. Finally, all participants engaged in a delay task and a second CPT designed to measure post-manipulation pain tolerance and subjective pain intensity levels.

**Results:** MS condition cohorts showed greater pain tolerance than controls on the post-manipulation CPT, though pain intensity levels did not differ between groups. Moderator analyses indicated that the relationship between the behavior perseverance facet of pain resilience and pain tolerance was significantly stronger among MS condition participants than controls.

**Conclusions:** This experiment is the first to document potential causal effects of MS on pain tolerance and Ms as a moderator of the association between self-reported behavior perseverance and behavioral pain tolerance. Findings provide foundations for extensions within clinical pain samples.

Corresponding author
Todd Jackson,
toddjackson@um.edu.mo

## INTRODUCTION

Pain is a complex, multifaceted experience that profoundly affects quality of life and daily functioning (*Ehde et al., 2003*; *Martz et al., 2005*). Understanding nuances in the interplay of psychosocial experiences with pain perception is vital for effective pain management (*Min et al., 2014*; *Turk, 2005*; *Turner et al., 2001*). Recent research has identified individual differences in pain resilience as a key influence upon how well people adapt to pain. According to *Slepian et al. (2016)* pain resilience has two facets: (i) behavioral and motivational tenacity (*i.e.*, behavioral perseverance) in the face of severe or prolonged pain and (ii) the perceived capacity to maintain a positive outlook in regulating emotions and cognition (*i.e.*, cognitive/emotional positivity) despite pain. Higher scores on these pain resilience dimensions have been linked to lower scores on measures of adverse outcomes including pain catastrophizing, fear of movement, impairment, pain-related anxiety and depression as well as elevations on measures of adaptive functioning such as general resilience to adversity, pain self-efficacy, hope, and optimism (*Ankawi et al., 2017b*; *Slepian et al., 2016*; *You & Jackson, 2021*). Notably, however, pain resilience has had mixed associations with pain intensity and tolerance in laboratory pain tasks such as the cold pressor test (CPT), as significant associations have been observed in some samples (*e.g.*, *Li & Jackson, 2020*; *Ling, Chen & Jackson, 2021*; *Slepian et al., 2016*) but not others (*e.g.*, *Ankawi, Slepian & France, 2015*; *Ankawi et al., 2020*). Given the somewhat inconsistent relations between pain resilience and pain perception, it may be useful to consider potential moderating factors that help to explain significant relations under some conditions but not others.

In this regard, terror management theory (TMT) offers a novel perspective on possible influences on pain perception and its links to individual difference influences such as pain resilience. Terror management theorists contend that awareness of mortality influences human thought, motivation, behavior, and emotion (*Greenberg, Pyszczynski & Solomon, 1986*; *Pyszczynski, Solomon & Greenberg, 2015*). Heightened awareness of death or mortality salience (MS) is hypothesized to bolster psychological defenses (*Burke, Martens & Faucher, 2010*) in a manner that may improve adaptive functioning. Specifically, people manage anxiety arising from thoughts about their death through (i) embracing an internalized version of their cultural worldview that provides explanations for origins and purposes of human life and transcendence beyond death and (ii) bolstering the sense that they are successfully living up to standards prescribed by such worldviews (self-esteem) (*e.g.*, *Pyszczynski, Solomon & Greenberg, 2015*). As such, accessibility to death-related thoughts should provoke increased worldview and self-esteem defenses and striving. Overall support for these contentions has been provided by a meta-analysis of 277 experiments that found MS manipulations yielded moderate effects ($r = 0.35$) on various worldview- and self-esteem-related dependent measures (*Burke, Martens & Faucher, 2010*).

In light of such data, MS may have utility in increasing the capacity to bear pain when resilience and/or acceptance of pain and suffering are emphasized within overarching cultural worldviews.

In East Asian cultures, the perception and management of pain and suffering are deeply influenced by Confucianism, Stoicism, and Buddhism (*Chen et al., 2008*; *Tung & Li, 2015*), which view pain and suffering as essential aspects of the human experience (*e.g.*, *Wei-Ming, 1984*) that contribute to moral self-realization (*Narayan, 2010*), strength of character (*Narayan, 2010*), and spiritual growth (*e.g.*, *Chen et al., 2008*). These cultural worldviews encourage enduring pain without outwardly expressing distress, aligning with values of personal resilience and social harmony (*Chen et al., 2008*). Consequently, pain is often managed privately, with stoic endurance seen as a virtue, reflecting a broader cultural acceptance of suffering as a path to personal and spiritual development (*Chen et al., 2008*; *Wang & Tian, 2018*). In tandem, these Chinese worldviews underscore cultural beliefs about the inevitability of pain and suffering, potential benefits of such experiences for personal growth and transcendence, and expectations that pain should be endured without distress, if possible, as a means of demonstrating strength of character and maintaining social harmony. *Ji et al. (2021)* found preliminary empirical support for a somewhat distinct Chinese worldview of pain and suffering that contrasted with a Euro-Canadian perspective. Across two studies, these authors found that Chinese participants (i) generated relatively more positive (or less negative) associations in response to the construct of "suffering" and (ii) added a greater number of positive ingredients and fewer negative ingredients in a hypothetical potion they created to represent what people experience while suffering compared to their Euro-Canadian peers. From a terror management perspective, increasing MS may foster awareness and the adoption of culturally-prescribed worldviews of how pain should be appraised and managed.

Select research has found preliminary support for links between MS manipulations and measures of pain perception. In particular, *McCabe, Carpenter & Arndt (2015)* assessed effects of MS (*vs.* a control condition) and false feedback linking pain endurance to heroic traits such as bravery, courage, and overcoming adversity (*vs.* certain positive personality traits) on pain reported from a cold pressor test (CPT) in a sample of U.S. men. Main effect analyses revealed exposure to the MS condition and false feedback linking pain endurance to heroism were associated with significantly less reported pain. These main effects were qualified by a significant interaction whereby MS condition men exposed to the heroic depiction of pain endurance reported significantly less pain than their peers in other conditions did. Although findings suggested that MS in tandem with depictions of bravery and resilience in the face of pain predict decreases in reported pain, the impact of these manipulations on objective measures of behavioral pain tolerance (*i.e.*, total time immersed in the CPT) was not assessed; examining effects on pain tolerance has implications for outcomes reflecting the capacity to function despite ongoing pain such as pain-related disability. Moreover, because the study was limited to U.S. men, it was not clear whether findings also applied to women or other cultural groups whose worldviews are characterized by acceptance and resilience in the face of suffering. Finally, despite the interaction of MS with a (heroic) false feedback manipulation having clear conceptual relations to resilience, it is not clear whether interactive effects of individual differences in specific dimensions of trait-related pain resilience (*i.e.*, behavioral perseverance and/or cognitive/affective positivity) with MS also influence pain tolerance or intensity.

To address these gaps, we investigated effects of an MS manipulation on tolerance and intensity of laboratory pain within a cultural group (*i.e.*, Chinese women and men) for whom dominant worldviews highlight pain and suffering as inevitable, potentially positive experiences to be endured without overt distress, in part, to demonstrate strength of character. We also assessed moderating effects of this manipulation on associations of perseverance and positivity dimensions of trait pain resilience with pain tolerance and intensity. Based on the preceding review, we hypothesized that higher scores on pain resilience dimensions of behavioral perseverance and cognitive/affective positivity as well as exposure to an MS manipulation (*vs.* a control manipulation) would be related to increased pain tolerance and lower levels of reported pain. Furthermore, based on preliminary evidence from *McCabe, Carpenter & Arndt (2015)*, we hypothesized that the manipulation would moderate relations between pain resilience dimensions and measures of pain tolerance and intensity.

## MATERIALS AND METHODS

### Research design and data collection procedures

We employed a randomized experimental design approved by the Human Research Ethics Committee of the associated university (Approval (H20071)), adhering to the Declaration of Helsinki. Recruitment focused on university students because this group has been found previously to exhibit more distinct responses to MS manipulations than non-students do (*Burke, Martens & Faucher, 2010*). Based on past research (*e.g.*, *Jackson et al., 2005*), exclusion criteria included the presence of a neurological disorder, serious mental illness (*e.g.*, bipolar disorder, schizophrenia), a current or past pain condition, a history of medical conditions including diabetes, Raynaud's disease, a circulation or cardiovascular disorder, anemia, hypertension, blood coagulation disorder, epilepsy, skin diseases or a past severe cold injury (*e.g.*, frostbite) as well as current medication use for any of these conditions. We also excluded people who had previously undertaken a CPT to control for effects of familiarity with experimental stimuli (*Wang, Jackson & Cai, 2016*). A gender-balanced sample of 80–92 women and 80–92 men was sought so that findings would apply across men and women. Because the quota of women was recruited more quickly, later stages of the recruitment process targeted men exclusively. This strategy yielded a closely balanced gender distribution for the final cohort.

Upon arrival, participants were informed about the general study focus (factors that might influence pain perception) and procedures (completion of several questionnaires and a CPT) as well as the time involved (35–45 min). After signing the informed consent and completing a checklist of exclusion criteria, participants engaged in a standardized 15 s practice CPT and completed self-report measures of demographics and pain resilience. Subsequently, they engaged in the (MS *vs.* control) experimental manipulation, a delay task, and a longer actual CPT, each of which is described below. Following the actual CPT, pain intensity ratings were solicited and participants were asked to guess the specific research purpose(s). The study was conducted from October 2020 to January 2021.

*Apparatus.* The CPT was conducted using a Model DX-208 cold water bath, measuring 25 cm × 25 cm × 20 cm, filled with 12.5 L of water at 2 °C (±0.1 °C). This temperature was

consistently maintained using a thermostat-regulated electric pump (*Wang, Jackson & Cai, 2016*). The CPT is widely used because it mimics effects of chronic pain conditions effectively due to its' unpleasantness and excellent reliability and validity (*e.g.*, *Jackson et al., 2005*; *Mitchell, MacDonald & Brodie, 2004*).

*Practice CPT.* Following standard published protocols (*Jackson & Phillips, 2011*; *Wang, Jackson & Cai, 2016*), participants first immersed their non-writing hand in room-temperature water for 30 s, followed by a 15-s immersion in 2 °C water. The practice CPT was used to ensure all those who engaged in the subsequent CPT were familiar with the experimental pain stimulus and had the same minimal baseline pain tolerance level prior to experimental manipulations.

## Completion of background measures

*Demographics.* Sex, age, height (centimeters), weight (kilograms), ethnicity, religion, relationship status (single, non-single), number of dependents, and total years of university education were assessed.

*Pain Resilience Scale-Chinese* (*You & Jackson, 2021*). The 10-item Chinese version of the Pain Resilience Scale (PRS; *Ankawi, Slepian & France, 2017a*; *Slepian et al., 2016*) was used to evaluate behavioral perseverance and cognitive/affective positivity facets of pain resilience in respondents. Items were rated on a scale ranging from 0 (not at all) to four (all the time). The Chinese PRS replicated the two-factor structure of the original PRS and has demonstrated reliability and validity in Chinese samples (*You & Jackson, 2021*). In this experiment, behavioral perseverance and cognitive/affective positivity subscales each had Cronbach's alpha values of $\alpha = 0.82$.

## Exposure to experimental manipulations

Participants were randomly assigned to either an MS group or a neutral control group. Those in the MS group responded to two standard open-ended questions designed to evoke thoughts about death: (a) "What will happen to you physically when you die?" and (b) "What emotions are aroused in you when you think about your death?" based on past work (*Rosenblatt et al., 1989*). Additionally, MS group members completed the 17-item University Student Personal Death Anxiety Scale (*Zhou et al., 2019*) as another way of increasing exposure to MS. Conversely, control group cohorts responded to two open-ended questions about personal reactions to television-viewing: (a) "What will happen to you when you watch television?" and (b) "What emotions are aroused in you when you think about watching television?" (*Greenberg et al., 1992*). Control group members also completed a 17-item scale related to satisfaction with television viewing (*Song, 2018*). Across these conditions, participants were instructed to engage in the writing task for 10 min (*Burke, Martens & Faucher, 2010*). Following experimental manipulations, all participants responded to three manipulation check items used to assess the validity of MS manipulations in other published research (*Guan et al., 2020*). Specifically, participants were asked, "How much did you think about death?" (2) "How much fear did you feel?" (3) "How unpleasant did you feel?" during the task on 11-point scales with "0 = not at all" and "10 = very strong" as anchors.

### Delay tasks

TMT posits that the effects of MS are most potent when thoughts of death are accessible yet not in conscious awareness, necessitating a short delay between MS inductions and responding to dependent measures for optimal impact (*Arndt, Greenberg & Cook, 2002*; *Greenberg et al., 2000*). Longer delays (7–20 min) and engagement in two or three different tasks during delays result in more significant MS effects than shorter delays (2–6 min), single task delays, or no delays do (*Burke, Martens & Faucher, 2010*). Accordingly, our experiment incorporated two distinct delay tasks. First, participants completed the Positive Affect and Negative Affect Scale (PANAS; *Qiu, Zheng & Wang, 2008*) which is commonly used as a delay task (*Burke, Martens & Faucher, 2010*). The PANAS-Chinese version comprises 18 items that evaluate positive and negative emotions experienced over the past week. Items were rated on a frequency scale ranging from one ("never or rarely") to five ('very strong') (*Qiu, Zheng & Wang, 2008*). Subsequently, participants engaged in a Sudoku game. A total time of 10 min was allocated for both tasks regardless of participant completion speed to ensure standardization of the delay time and optimize potential effects of the MS manipulation.

### Actual CPT

For the actual CPT, participants were to immerse their left hand in cold water for as long as possible although they could withdraw at any point, particularly if the pain was unbearable. During the immersion, they could use any coping strategy they chose to manage the pain though the experimenter who quietly recorded its duration from behind would not engage with them until after the CPT was terminated. Unknown to participants, the maximum immersion time was 4 min, after which they were told to withdraw the hand if they reached the time limit.

### Measurement of post-CPT pain tolerance and pain intensity

Pain tolerance from the actual CPT was based on the duration each participant's hand remained immersed in ice water to the nearest hundredth of a second up to a 4-min time limit. Immediately after the CPT, participants answered three widely used pain intensity items (*Jackson et al., 2012*; *Wang, Jackson & Cai, 2016*) assessing pain intensity at the moment one withdrew from the ice water, as well as average pain intensity during the course of the CPT and highest level of pain experienced during the immersion. Each item was rated on a numeric scale with 0 ("no pain") and 10 ("worst pain imaginable") as anchors. Responses from these items were averaged to obtain total pain intensity scores. In this sample, the three-item pain intensity scale had an internal consistency of $\alpha = 0.90$.

### Debriefing

Following the second CPT, participants were asked to guess the specific research purposes and hypotheses as a means of assessing awareness of research questions as an influence on results. They were then informed of the main research focus, given the opportunity to ask lingering questions and paid 30 RMB and thanked for their time and participation.

## Data analyses

The sample size estimate was based, in part, on G*power 3.1 software. Power analysis estimated minimum of N's of 99 participants and 140 participants (70 per group), respectively, for multiple linear regression and t-test analyses, based on medium effect sizes ($f^2 = 0.15$ or $d = 0.50$) (*Cohen, 1992*), with 90% power and a 5% error probability. Based on these parameters, minimum sample size requirements were met. In addition, we sought a final sample size that approximated that of *McCabe, Carpenter & Arndt*'s *(2015)* conceptually-related experiment ($N = 160$) to ensure that statistically significant effects in the present experiment were not due to using a much larger sample size. SPSS 20.0 was employed for analyses. Independent samples t-tests and chi-square tests were used to assess MS *vs.* control group differences on measures of demographics, pain resilience dimensions and manipulation checks based on a significance threshold of $p < 0.05$. MS *vs.* control condition differences in pain tolerance and pain intensity were evaluated *via* analyses of covariance (ANCOVA), adjusting for potential differences on background characteristics.

We calculated Spearman correlation coefficients to identify statistically significant relations of experimental manipulation conditions and dimensions of pain resilience with dependent measures (pain tolerance and pain intensity). When the experimental manipulation had a significant effect on a dependent measure, moderator analyses were conducted using Jamovi (https://www.jamovi.org) and the Process macro in SPSS 20.0, supplemented by a 5,000-iteration bootstrapping procedure to generate model estimates and confidence intervals (CIs). This non-parametric approach was used to identify interaction and their statistical significance, defined by excluding zero in the bootstrapped 95% confidence intervals (*Preacher & Hayes, 2008*; *Shrout & Bolger, 2002*; *Tetreault et al., 2018*). Variables were standardized as z-scores prior to conducting the moderator analysis.

# RESULTS

## Preliminary analyses

From an initial sample of 182 healthy college students, data from five participants were excluded for failing to display minimal pain tolerance (*i.e.*, lasting less than 15 s) on the actual CPT based on other published work suggesting such responses are highly anomalous and reflect a lack of effort (*Jackson & Phillips, 2011*; *Jackson et al., 2009*; *Wang, Jackson & Cai, 2016*). Results were fully replicated when these data were included in main analyses. Data from seven other participants were also excluded for inadequate responses to the experimental manipulation (*i.e.*, answers to the two open-ended questions were overly brief and reflected a lack of engagement); manipulations did not have differential effects on this factor, $\chi^2 = 1.24$, $p = 0.266$, suggesting experimental conditions did not differ regarding overall engagement in completing MS *vs.* control condition tasks. Finally, none of the participants guessed the specific research purposes.

The final sample comprised 86 women and 84 men, primarily of Han Chinese ethnicity (84%), no formal religious affiliation (94%), and right-handedness (100%). A majority reported being single (62%). The sample had an average age of 19.74 years (SD = 1.53,

**Table 1 Mortality salience and control group differences on demographic measures (N = 170).**

| Characteristics measure | Mortality salience N = 85 M (SE) | Control N = 85 M (SE) | $\chi^2$ /t | p | Cohen's d |
|---|---|---|---|---|---|
| Gender (% male) | 49% | 49% | 0.00 | 1.000 | – |
| Ethnicity (% Han) | 84% | 84% | 0.00 | 1.000 | – |
| Religion status (% no) | 96% | 92% | 1.70 | 0.192 | – |
| Relationship status (% single) | 61% | 64% | 0.10 | 0.752 | – |
| Age | 19.51 (1.41) | 19.96 (1.62) | −1.97 | 0.051 | 0.30 |
| Body mass index | 21.07 (3.33) | 21.22 (2.55) | −0.34 | 0.733 | 0.05 |
| Years of university education | 2.08 (1.25) | 2.35 (1.37) | −1.35 | 0.180 | 0.21 |
| Number of dependents | 4.05 (1.22) | 3.86 (1.03) | 1.09 | 0.279 | 0.17 |

Notes:
Values are mean (SE) for continuous variables, $n$% for categorical variables.

range: 18–26 years), a mean of 2.22 years of university education (SD = 1.31 years, range = 1–7 years), an average of 3.95 dependents in their family (SD = 1.13, range = 1–8) and a mean body mass index of 21.15 (SD = 2.96, range = 15.24–35.83).

No significant MS *versus* control group differences were found on demographic measures or pain resilience (see Tables 1 and 2). However, because a statistical trend emerged for age (*i.e.*, the MS group was slightly younger than the control group, $p = 0.051$), age was included as a covariate in main analyses of group differences in pain tolerance and intensity to be conservative. Results were fully replicated when age was not treated as a covariate in main analyses. Regarding manipulation check items, MS group participants reported significantly more thoughts of death (MS group: M = 7.18, SD = 1.83 *vs.* control group: M = 0.42, SD = 1.36, t(168) = 27.29, $p < 0.001$), feelings of fear (MS group: M = 4.31, SD = 2.30 *vs.* control group: M = 0.66, SD = 1.25, t(168) = 12.83, $p < 0.001$) and feeling unpleasant (MS group: M = 4.19, SD = 2.48 *vs.* control group: M = 1.65, SD = 1.75, t(168) = 7.73, $p < 0.001$). Hence, experimental manipulations designed to ensure group differences in MS were effective.

## Main analyses

As shown in Table 2, the MS group exhibited significantly longer pain tolerance on the CPT than the control group did ($p = 0.001$) with a medium effect size strength (Cohen's d = 0.54). Conversely, there was no significant experimental condition difference in overall pain intensity ($p = 0.743$); the corresponding effect size was very small (Cohen's d = 0.05). As presented in Table 3, behavioral perseverance and cognitive/affective positivity facets of self-reported pain resilience had significant positive correlations with pain tolerance as well as negative correlations with pain intensity; related effect size magnitudes were small based on *Cohen (1992)*. In line with ANCOVA results, random assignment to the MS (*vs.* Control) manipulation had a significant positive correlation with pain tolerance and a non-significant association with pain intensity. In light of these bivariate correlations, we tested potential moderating effects of MS on relations of behavioral perseverance and

**Table 2 Mortality salience and control group differences on measures of pain resilience, pain tolerance and pain intensity (N = 170).**

| Characteristics measure | Mortality salience N = 85 M (SD) | Control N = 85 M (SD) | t/F | p | Cohen's d | Difference MS minus CG (95% CI) |
|---|---|---|---|---|---|---|
| Pain Resilience Scale–Chinese | 2.50 (0.58) | 2.38 (0.54) | 1.40 | 0.164 | 0.21 | 0.12 [−0.05 to 0.29] |
| Behavior perseverance | 2.70 (0.73) | 2.67 (0.62) | 0.26 | 0.792 | 0.04 | 0.03 [−0.18 to 0.23] |
| Cognitive/affective positivity | 2.43 (0.63) | 2.27 (0.60) | 1.69 | 0.094 | 0.26 | 0.16 [−0.03 to 0.35] |
| Pain tolerance | 84.16 (74.94) | 49.57 (50.58) | 11.74 | 0.001 | 0.54 | 34.59 [15.24 to 53.96] |
| Pain intensity | 6.77 (1.47) | 6.84 (1.26) | 0.11 | 0.743 | 0.05 | 0.07 [−0.49 to 0.34] |

Notes:
Pain tolerance and intensity differences are reported after first controlling for all other measures on which resilience subgroups had significant (or margin significant) differences in analyses of covariance (*i.e.*, age). MS, mortality salience; CG, control group; CI, confidence interval.

**Table 3 Correlations between facets of pain resilience, experimental manipulation, pain tolerance and pain intensity in the study sample.**

| | 1 | 2 | 3 | 4 | 5 |
|---|---|---|---|---|---|
| 1 Behavior perseverance | – | | | | |
| 2 Cognitive/Affective positivity | 0.456*** | | | | |
| 3 Experimental manipulation | 0.024 | 0.121 | | | |
| 4 Pain tolerance | 0.169* | 0.288*** | 0.360*** | | |
| 5 Pain intensity | −0.145 | −0.238** | 0.003 | −0.272*** | – |

Notes:
* $p < 0.05$
** $p < 0.01$
*** $p < 0.001$.

cognitive/affective positivity with pain tolerance while moderator analyses were not run for pain intensity as an outcome.

As highlighted in Table 4, random assignment to the MS condition moderated the relationship between behavioral perseverance and pain tolerance; the experimental manipulation × behavior perseverance interaction remained significant even when main effects of experimental manipulation condition and behavior perseverance were retained in the model. The 95% bias-corrected confidence interval for this interaction excluded zero, further underscoring the significant moderating effect of MS. Finally, in support of moderation, a simple slope analysis, conditioned at ±1 SD from the mean (*Preacher, Curran & Bauer, 2006*) found that self-reported behavioral perseverance and pain tolerance had significant association with a medium effect size magnitude in the MS group ($\beta_{simple} = 0.31$, SE = 0.10, $p = 0.001$) and a non-significant association in the control group, with a very small effect size ($\beta_{simple} = 0.00$, SE = 0.11, $p = 0.988$). In contrast to these results, the experimental manipulation did not significantly moderate the relationship between cognitive/affective positivity and pain tolerance (see Table 4).

## DISCUSSION

Building on separate lines of research that have identified experimental manipulations of MS (*e.g.*, *McCabe, Carpenter & Arndt, 2015*) and individual differences in trait measures of pain resilience (*Ankawi et al., 2017b*; *Li & Jackson, 2020*; *Slepian et al., 2016*; *You &*

**Table 4 Moderating effects of mortality salience on association of between pain resilience dimensions and pain tolerance in the study sample ($N = 170$).**

| Measure | $\beta$ | BootSE | $t$ | Boot LLCI | Boot ULCI |
|---|---|---|---|---|---|
| Behavior perseverance | 0.156 | 0.076 | 2.115* | 0.009 | 0.308 |
| Experimental manipulation | 0.259 | 0.072 | 3.573*** | 0.118 | 0.403 |
| Behavior perseverance × Experimental manipulation | 0.154 | 0.075 | 2.086* | 0.0001 | 0.295 |
| Overall Model $R^2 = 0.13$*** Overall Model F = 7.910*** | | | | | |
| Cognitive/Affective Positivity | 0.214 | 0.074 | 2.902** | 0.068 | 0.359 |
| Experimental manipulation | 0.235 | 0.074 | 3.196** | 0.090 | 0.380 |
| Cognitive/Affective Positivity × Experimental manipulation | 0.046 | 0.074 | 0.617 | −0.100 | 0.191 |
| Overall Model $R^2 = 0.12$*** Overall Model F = 7.325*** | | | | | |

Notes:
β, Standardized Beta Coefficient; Boot SE, Bootstrap Standard Error; LLCI, lower level for confidence interval; ULCI, upper level for confidence level.
* $p < 0.05$
** $p < 0.01$
*** $p < 0.001$.

*Jackson, 2021*) as potential influences on pain perception, our research evaluated the causal impact of MS on pain tolerance and intensity as well as its moderating effects on relations between pain resilience and pain perception. Analyses provided partial support for hypotheses. Associations between higher levels of self-reported pain resilience and longer behavioral pain tolerance replicated past several experimental pain studies (*e.g.*, *Li & Jackson, 2020*; *Ling, Chen & Jackson, 2021*; *Slepian et al., 2016*). Regarding more novel findings, this experiment is the first to document a significant association between exposure to an MS (*vs.* control condition) manipulation and longer tolerance of laboratory pain, even though these experimental conditions did not have significant differential effects on reported pain intensity levels. Furthermore, moderator analyses resulted in the novel finding that being randomly assigned to the MS condition was related to a significant moderate positive association between pre-task self-reported trait behavioral perseverance and objectively-measured behavioral pain tolerance while random assignment to the control condition resulted in a very small, non-significant association between these variables. Conversely, MS did not moderate cognitive/affective positivity-pain tolerance relations. Implications of novel MS and moderator analysis findings are elaborated briefly below.

An overarching premise of TMT is the view that heightened awareness of death can facilitate adaptive outcomes (*Burke, Martens & Faucher, 2010*). We tested this contention based on responses to painful laboratory stimulation within a young Chinese adult sample. Selective support was found for this perspective, as MS condition participants demonstrated greater behavioral tolerance for cold pressor pain than control condition cohorts did. This effect was especially notable because no MS *vs.* control condition difference was observed for overall pain intensity. As such, the significantly stronger capacity to endure painful stimulation displayed among MS condition participants was not due to experiencing comparatively less severe pain. Furthermore, because participants

were randomly assigned to MS *vs.* control condition manipulations, group differences on measures of background functioning were also unlikely to account for this difference, at least in theory. As such, our findings suggest that exposure to MS cues has a significant causal impact on the capacity to bear painful stimulation for more extended periods of time.

The absence of a significant MS *vs.* control condition difference in reported pain intensity aligns with results from a small ($N = 18$) college student sample of Chinese men (*Wang & Tian, 2018*) whereby pain intensity ratings did not differ between an MS priming condition on 1 day and a control priming condition on the second day. On the surface, null effects in Chinese samples appear to diverge from elements of Chinese worldviews reflecting expectations that pain should be endured without showing emotion (*Wang & Tian, 2018*). Strictly speaking, however, sensory pain indexes such as subjective intensity ratings are not synonymous with affective measures that tap pain unpleasantness or negative emotional reactions to painful stimulation such as pain catastrophizing. Hence, because overt expressions of emotion were not assessed in China-based experiments, MS *vs.* control group differences in subjective pain ratings may have been attenuated. Null effects on pain intensity from the present study also contrast with evidence from *McCabe, Carpenter & Arndt (2015)* who found reminders of mortality (*vs.* a control topic) resulted in lower pain intensity and unpleasantness ratings in a U.S. sample. Possible differences in cultural worldviews (*e.g.*, norms related to overt expressions of emotion) and the exclusion *vs.* inclusion of "unpleasantness" in the measurement of reported pain may help to explain these discrepancies.

In sum, main effect results for the MS *vs.* control condition manipulation suggested that procedures used to induce MS have potential causal effects on the capacity to endure laboratory pain, independent of subjective pain intensity levels. As such, these findings provide experimental foundations for extensions of relevant theoretical frameworks such as existential psychotherapy and intervention strategies designed to increase MS within future pain management studies of laboratory pain, acute pain, and chronic pain. In a related meta-analysis on the efficacy of existential therapies, *Vos, Craig & Cooper (2015)* concluded that the overall quality of intervention studies warrants improvements but structured interventions incorporating facets of existential psychotherapy related to mortality and meaning can have direct, positive effects on physically ill patients. Furthermore, in line with our MS (*vs.* control) manipulation effects upon pain tolerance but not subjective pain intensity, *Gebler & Maercker (2014)* found a cognitive behavioral therapy (CBT) intervention that incorporated tenets of an existential perspective led to significant post-treatment reductions in impaired daily functioning despite pain but no difference in subjective pain severity compared to CBT-alone.

Moderator analyses underscored a significant correlation of a medium effect size strength ($\beta = 0.31$) between pre-CPT self-reports of trait behavioral perseverance and behavioral tolerance of cold pressor pain among participants exposed to the MS manipulation. Conversely, the behavioral perseverance-pain tolerance correlation had a very small effect size strength ($\beta = 0.00$) in the control condition. These results have parallels with moderator analyses from *McCabe, Carpenter & Arndt (2015)* who observed a

significantly lower mean reported pain rating among men exposed to an MS manipulation and a false feedback manipulation linking pain endurance to heroic depictions that, in part, reflected resilience (*e.g.*, overcoming adversity) compared to men in other experimental conditions. Several other experiments have also found exposure to MS manipulations may bolster strengths of relation between particular self-regulatory behaviors and responses on self-report measures of related constructs such as self-control and desire for control (*e.g.*, *Alper & Ozkan, 2015*; *Kelley & Schmeichel, 2015*; *Kelley, Tang & Schmeichel, 2014*). Essentially, this small body of research suggests that MS manipulations could act as a psychological catalyst that enhance relations of positive self-perceptions with related behavior responses. Significant moderating effects observed in the present experiment are preliminary and need to be replicated but also suggest that mortality reminders may increase the capacity to bear pain, particularly among people who are already endowed with strong beliefs that they can persevere in their daily tasks despite experiences of pain.

Finally, elevations on the cognitive/affective positivity dimension of pain resilience also had a significant positive correlation with behavioral tolerance of cold pressor pain yet the MS manipulation did not moderate the association of reported positivity levels with behavioral pain tolerance times. Given that cognitive/affective positivity reflects the capacity to experience positive emotions and maintain an optimistic outlook despite pain, it is possible that MS may have moderating effects on more directly relevant outcomes such as state optimism or positive affect during exposure to painful stimulation instead of less conceptually relevant outcomes such as tolerance of cold pressor pain. This conjecture should also be a focus of future studies.

## STRENGTHS AND LIMITATIONS

Given the growing recognition of psychological factors as important influences on the experience and management of pain, our focus on exposure to an MS manipulation as a potential cause and moderator of responses to painful stimulation is a novel aspect of this research. The use of a large, mixed gender sample and experimental study design featuring random assignment to carefully matched experimental manipulations that permitted evaluations of possible causal effects of MS *vs.* control conditions on pain tolerance and intensity were related methodological strengths that provide empirical foundations for related tests within acute pain and chronic pain samples.

The main limitations of this study also merit attention. First, although the assessment of college students was useful because this population may be especially sensitive to effects of MS manipulations (*Burke, Martens & Faucher, 2010*), findings may not generalize to clinical pain samples, other age groups or different socioeconomic status groups. Second, despite support for the hypothesis that the MS manipulation would increase tolerance for a particular laboratory stimulus of a brief duration (cold), it is not clear whether MS manipulations influence the capacity to bear pain over extended intervals or apply to other kinds of noxious stimulation. Third, although results underscored effects of MS (*vs.* control condition) manipulations on laboratory pain and their relations to specific facets of pain resilience, we could not directly test whether increases in defenses reflecting Chinese

cultural worldviews and self-esteem were the specific mechanisms that explained pain tolerance results, in part, because there are no clear guidelines for how or when to evaluate these defenses within laboratory pain paradigms. The use of free association strategies in response to "suffering" (*e.g.*, *Ji et al., 2021*) or "pain" warrants consideration as a means of accessing TMT defenses such as cultural worldviews (*Burke, Martens & Faucher, 2010*) *vs.* other alternate factors such as changes in appraisals of pain as a threat or a challenge (*e.g.*, *Jackson, Wang & Fan, 2014*) as mechanisms that account for MS manipulation effects on behavioral pain tolerance. Finally, random assignment to distinct standardized manipulations is a widely accepted means of controlling for unwanted sensitization effects and group differences on innumerable background factors that are simply not feasible to measure. However, random assignment is not a panacea. Replications are needed to ensure causal effects of MS manipulations in this experiment are robust across independent samples.

## CONCLUSIONS

In conclusion, this experiment is the first to document causal effects of an MS manipulation on tolerance for cold pressor pain and its role as a moderator of the association between the self-reported behavioral perseverance and behavioral pain tolerance. Exposure to reminders of death resulted in significantly increased pain tolerance and a significantly stronger positive correlation between pre-task beliefs about behavioral perseverance capacities and actual pain tolerance relative to exposure a control manipulation. These findings offer compelling, initial empirical evidence for contemplation of mortality as a facilitative influence on pain tolerance, especially among people who already have strong beliefs in their capacity to persevere in daily tasks despite pain. Replications and extensions are needed to evaluate the stability of these findings and gauge their relevance and applicability to clinical pain samples.

## ACKNOWLEDGEMENTS

The authors acknowledge all participants who helped to facilitate data collection.

### Funding

This work was supported by the Science and Technology Fund Project of Guizhou Provincial Health Commission (gzwkj2022-482), Special Research Project in the Nursing Discipline at Guizhou Medical University (YJ22015), High-Level Talent Startup Fund Project at Guizhou Medical University (Doctoral Contract J2021057), Chinese National Natural Science Foundation Incubation Program at Guizhou Medical University (22NSFCP41) and Chinese National Natural Science Foundation (31871141). There was no additional external funding received for this study. The funders had no role in study design, data collection and analysis, decision to publish, or preparation of the manuscript.

## Grant Disclosures

The following grant information was disclosed by the authors:

Science and Technology Fund Project of Guizhou Provincial Health Commission: gzwkj2022-482.

Special Research Project in the Nursing Discipline at Guizhou Medical University: YJ22015.

High-Level Talent Startup Fund Project at Guizhou Medical University: J2021057.

Chinese National Natural Science Foundation Incubation Program at Guizhou Medical University: 22NSFCP41.

Chinese National Natural Science Foundation: 31871141.

## Competing Interests

The authors declare that they have no competing interests.

## Author Contributions

- Beibei You conceived and designed the experiments, performed the experiments, analyzed the data, prepared figures and/or tables, authored or reviewed drafts of the article, and approved the final draft.
- Hongwei Wen analyzed the data, prepared figures and/or tables, authored or reviewed drafts of the article, and approved the final draft.
- Todd Jackson conceived and designed the experiments, prepared figures and/or tables, authored or reviewed drafts of the article, and approved the final draft.

## Human Ethics

The following information was supplied relating to ethical approvals (*i.e.*, approving body and any reference numbers):

The Human Research Ethics Committee of Southwest University granted Ethical approval to carry out the study (Ethical Application Ref: H20071).

## Data Availability

The raw measurements are available in the Supplemental File.

## Supplemental Information

Supplemental information for this article can be found online at http://dx.doi.org/10.7717/peerj.17204#supplemental-information.

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
