# Peer review of "Investigating mortality salience as a potential causal influence and moderator of responses to laboratory pain"

_PeerJ, doi:10.7717/peerj.17204_

## Round 0.1 · original submission · Major Revisions

Thank you for submitting your manuscript to PeerJ. We apologize for the delay in our review process. The somewhat contradictory evaluations from our initial two reviewers necessitated the involvement of a third reviewer. Before I present their individual feedback, it is essential to note that your manuscript will require substantial revisions before it can be considered for publication.

Reviewer 1 has expressed a need for more detailed information on your experimental and statistical methods, a viewpoint I share. Furthermore, Reviewer 3 has highlighted significant concerns regarding your methodology and statistical analysis. Thoroughly addressing all the issues raised by the reviewers will undoubtedly improve the quality of your manuscript.

Reviewer 1 ·

Basic reporting

The present study, which elucidates that the incorporation of MS contributes to the prolongation of pain tolerance, is indeed intriguing. However, it would be beneficial for the clarity and coherence of the study if a substantial reorganization of both the Introduction and Methods sections is undertaken. Specifically, simplifying the Introduction while concurrently concentrating the review on the concepts deployed in this experiment would not only streamline the narrative but also enhance its persuasive power. Kindly reconsider the structure of your paragraphs and pay meticulous attention to the crafting of each paragraph to ensure clarity and cogency in your exposition.

Experimental design

It is imperative to review past studies that utilized the cold pressor test in the measurement of pain resilience in the Introduction section of the paper. The correspondence between the questionnaire employed in the study and the concepts or reviews introduced in the Introduction is not clearly delineated.

Validity of the findings

For some analyses, the reason for using the method should be clearly stated.

Additional comments

It is advisable to have your manuscript reviewed for language accuracy by a fluent English speaker.

Annotated reviews are not available for download in order to protect the identity of reviewers who chose to remain anonymous.

Reviewer 2 ·

Basic reporting

Fine.

Experimental design

Fine.

Validity of the findings

Fine.

Study is not particularly exciting... This study represents a small scale experiement to demonstrate that priming people with death appears to increase their pain tolerance somewhat on a fairly arbirtrary task (the CPT).

It hasn't been made abundantly clear in the article what the magnitude of any effects are.

Reviewer 3 ·

Basic reporting

There are instances where language should be improved and/or typographical errors removed. For example, lines 110-111, please use the correct (past) tense, line 217, correct typo healthy, line 246, I am wondering if this should be in the present tense instead, and line 304, subject missing. I suggest you review the manuscript or contact a professional editing service.

Line 49-50, this needs a reference.

Line 60, Perry et al. has no date.

Line 60, the acronym MS needs to be introduced first.

I think overall the introduction could benefit from a more in-depth explanation of MS, the relationship between MS, resilience and pain, and discussion of the potential impact of studying this phenomenon in a pain-context (i.e., how would this contribute to the development of more effective pain management approaches?).

Line 92, use the acronym MS consistently throughout.

Line 135, I think this paragraph could be improved by clarifying that you, in fact, used the PRS-C, as it is currently difficult to discern whether a (novel) 10-item scale for behavioural perseverance and cognitive/affective positivity specifically or the PRS-C was used.

For the control manipulation, could you report the exact questions?

Line 165 – 169: “Evidence has shown that participants who experienced two or three tasks during the delay demonstrated more significant effects compared to those who engaged in only one task. Likewise, longer delays (7–20 min) produced more pronounced effects than shorter delays (2–6 min) or no delays at all (Burke et al. 2010).”, What exactly do you mean by experiencing two or three tasks?
Line 173 – 174, is this ten minutes for each task for a total of 20 minutes? In which order were the tasks given?

Experimental design

Please can you clarify whether there was an exclusion criterion that specified an upper age limit of 26 and if so, why this was employed. With age being an insignificant factor, an age limit seems redundant.
Did recruit specifically to have an equal number of men and women in the study? If so, please report this and report how participants were recruited.

Did you check whether participants had ingested pain medication prior to taking part and whether participants suffered from acute or chronic pain? These factors may have influenced pain intensity ratings and pain tolerance.

I am concerned about the study sequence, specifically that participants completed the questionnaires after the first cold pressor immersion, i.e., after experiencing pain. Could this have affected scores on the questionnaires?

How many participants were excluded due to guessing the purpose of the study and how was this operationalised?

Can you confirm that pain intensity was derived by averaging the three ratings?

What is your reasoning behind selecting a medium effect size for your power calculation? Was this based on previous research? I am also wondering whether the power calculation should be for the moderation analysis, rather than the ANCOVA?

Validity of the findings

I do not follow your reasoning behind excluding participants who fail to keep their hand in the cold pressor for more than 15 seconds. Considering the instructions were to keep the hand immersed for as long as possible, this should be considered a viable outcome. I would suggest including these participants in the analyses. Similarly, please explain why seven participants were excluded due to overly brief responses to the open-ended questions. How was this operationalised and how did you decide to include or exclude participants?

It seems rather odd that you would include age as a covariate in the ANCOVA going against the same logic applied to the other covariates. It would seem more sensible to be consistent in the argument for inclusion or exclusion of covariates and, consequently, exclude age from the analysis.

Line 237, you mention confirming the success of the MS-induction. I disagree with this, as you cannot draw this conclusion from the data. Differences pertaining to MS may have been present before the induction. Since you did not assess MS before and after the induction, you cannot conclude that the MS-induction was successful, but only that there was a significant difference in terms of MS between the groups. This should be made clear in the text.

I am curious as to why you did not correct for baseline pain intensity in your analysis? Since you did not exclude participants who suffer from acute or chronic pain, there may be significant variations in pain levels and so the pain intensity reported upon removal of the hand from the water may be biased by differences in baseline pain levels. I would suggest, if possible, to subtract baseline pain levels from post immersion pain levels to have a baseline-corrected pain intensity measure.

Additional comments

Thank you for the opportunity to review the manuscript titled: “Investigating Mortality Salience: Causal Effects on Pain Tolerance and Its Moderating Role on the Association Between Pain Resilience and Pain Tolerance” by You and Jackson. The study covers an interesting and relevant phenomenon in mortality salience and represents research relevant for this journal. Nevertheless, I am concerned about some of the analytical choices, especially in the selection of covariates and the exclusion of participants, and the lack of detail required for replication, particularly considering this study was not pre-registered.

Have you considered examining pain sensitivity?

Line 252, when you say “MS manipulation”, do you mean the assigned group or the MS value?
In your introduction you said you would examine whether MS moderates the relationship between pain resilience and pain tolerance, yet you look at this separately for behavioural perseverance, cognitive/affective positivity. Could you explain this disparity?

Line 299, “Those under MS might be engaging these defenses, provoked by contemplation of death, to withstand pain. “, please can you expand on this? I am still not clear on how contemplation of death and the activated defences motivates individuals to withstand pain.

Line 341-345, “Maxfield et al. (2012) demonstrated that older adults with high levels of executive functioning (i.e., positively oriented adaptation to challenges) responded to MS with increased tolerance towards moral transgressors, whereas those with diminished executive functioning adopted a more punitive stance (Maxfield et al. 2012).”, I do not entirely see the relevance here. This entire paragraph seems rather speculative. What is the relevance of moral transgressors here? Overall, you need to relate this paragraph more to pain tolerance and how these might be related.

Line 406, please can you explain why your experimental study is considered cross-sectional?

---

## Round 0.2 · Minor Revisions

Thank you for submitting the revised manuscript. I sent this to the two reviewers who engaged in the previous round. Both reviewers are mostly satisfied with your revisions but some minor concerns remain. Thus, the decision is "minor revisions." For the details, please check the reviewers' comments.

Reviewer 1 ·

Basic reporting

Thank you for revising the manuscript. It has become more readable than before. I notice some discrepancies between the introduction and the discussion sections. Therefore, I recommend adjusting the introduction to align with the structure of the discussion.

Experimental design

The experimental design has become clearer with the revision, particularly the analysis section, which I find significantly improved. While the importance of the questionnaires used was well understood in the objectives, their presence feels somewhat diminished in the discussion. It's not immediately clear why they were essential and how they contribute to the interpretation of the results. Could this be improved?

Validity of the findings

This is a fascinating study, and I believe there is no need to overly emphasize race or culture in the introduction.

Annotated reviews are not available for download in order to protect the identity of reviewers who chose to remain anonymous.

Reviewer 3 ·

Basic reporting

The authors have greatly improved the manuscript in this regard and all my concerns have been addressed.

Experimental design

Thank you for taking the time to respond to my queries. I think the changes you have implemented have greatly improved the clarify of the manuscript and the methods employed.

Validity of the findings

All points raised have been addressed.

Additional comments

The authors have done an excellent job responding to the comments and the manuscript has been improved significantly. It does appear, however, as though an additional author was added to the manuscript. I cannot find any explanation as to the reasoning behind this addition. Please, could it briefly be made clear why a third author was added? Perhaps include a short paragraph about author contributions to clarify the issue.

---

## Round 0.3 · accepted · Accept

Because the decision in the previous round was "minor revision", I checked the revised manuscript without sending the reviewers. I believe that the manuscript should be improved.